# MOdern River archivEs of Particulate Organic Carbon: MOREPOC

Yutian Ke[1*], Damien Calmels[1], Julien Bouchez[2], Cécile Quantin[1]

[1] GEOPS, Université Paris-Saclay-CNRS, Orsay, 91405, France

[2] Université de Paris-Cité, Institut de physique du globe de Paris, CNRS, Paris, 75005, France

[*] Now at Division of Geological and Planetary Sciences, California Institute of Technology, Pasadena, CA 91125, USA

*Correspondence to*: Yutian Ke (yutianke@caltech.edu; yutian.ke@universite-paris-saclay.fr)

**Abstract.** Riverine transport of particulate organic carbon (POC) associated with terrigenous solids to the ocean has an important role in the global carbon cycle. To advance our understanding of the source, transport, and fate of fluvial POC from regional to global scales, databases of riverine POC are needed, including elemental and isotope composition data from

contrasted river basins in terms of geomorphology, lithology, climate, and anthropogenic pressure. Here, we present a new, open-access, georeferenced, global database called Modern River Archives of Particulate Organic Carbon (MOREPOC) version 1.1, featuring data on POC in suspended particulate matter (SPM) collected at 233 locations across 121 major river systems. This database includes 3,546 SPM data entries, among which 3,053 with POC content, 3,402 with stable carbon isotope ($\delta^{13}C$) values, 2,283 with radiocarbon activity ($\Delta^{14}C$) values, 1,936 with total nitrogen content, and 299 with aluminum-

to-silicon ratio (Al/Si). The MOREPOC database aims at being used by the Earth System community to build comprehensive and quantitative models for the mobilization, alteration, and fate of terrestrial POC. The database is made available on the Zenodo repository in machine-readable formats as data table and GIS shapefile at https://doi.org/10.5281/zenodo.7055970 (Ke et al., 2022).

## 1. Introduction

Rivers are the main conveyor of terrestrial material to the ocean in the form of suspended particulate matter (SPM), which carries particulate organic carbon (POC) (Leithold et al., 2016; Blair and Aller, 2012). POC is defined as the fraction of total organic carbon contained in the solid fraction recovered after filtration of river water. Before reaching coastal environment and being eventually buried at the ocean bottom, terrestrial POC may experience alteration and/or degradation processes during fluvial transport. These processes need to be better quantified as they are key features of the global carbon cycle, particularly

in the context of current global environmental changes.

Riverine POC is a mixture of organic carbon (OC or $C_{org}$) from various sources, which can be split into two major origins: biospheric POC ($POC_{bio}$) and petrogenic POC ($POC_{petro}$) (Blair et al., 2003; 2004; Galy et al., 2007; Hilton et al., 2008). Land plants, soils, aquatic organisms, and microbes can all contribute radiocarbon-active riverine POC, with ages ranging from modern to multi-millennial (Galy et al 2007; Blair et al., 2010; Hilton et al., 2011). Radiocarbon-dead $POC_{petro}$ is derived from

the erosion of sedimentary rocks and consists of terrestrial or marine organic carbon photosynthesized millions of years ago

and has survived to at least one full erosion/sedimentation/exhumation cycle (Galy et al., 2008a; Hilton et al., 2011). The balance between the release of $CO_2$ by oxidation of $POC_{petro}$ and the drawdown of $CO_2$ by burial of $POC_{bio}$ in marine sediments controls the impact of the OC cycle on atmospheric $CO_2$ level over geological timescales (> 100,000 years). The resulting long-term global carbon fluxes are similar in magnitude to those from silicate weathering and volcanism (Berner, 2003; Hilton et al., 2014; Petsch, 2014; Galy et al., 2007; Galy and Eglinton, 2011; Hilton and West, 2020). Net continental $POC_{bio}$ burial accounts for about 35-70 MtC/yr considering that only 30% of the total riverine input to the ocean is efficiently buried (Blair and Aller, 2012; Burdige, 2005; Galy et al, 2015), while the oxidation of $POC_{petro}$ in sedimentary rocks would release about 40-100 MtC/yr to the atmosphere (Petsch, 2014; Hilton and West, 2020). These fluxes are comparable to those induced by silicate weathering, carbonate weathering by oxidation of sulfides, and volcanism, demonstrating that POC could play an important role in the Earth's long term carbon cycle (Berner, 2003; Hilton et al., 2014; Petsch, 2014; Galy et al., 2007; Galy and Eglinton, 2011; Hilton and West, 2020). Consequently, it is fundamental to quantify POC sources and fluxes as well as to understand the fate of the different POC pools, in order to better constrain the role that POC plays in the global carbon cycle. To that aim, radiocarbon activity provides unique information on POC age, residence time, and source. Thanks to improved carbon-dating technology and more easily accessible accelerator mass spectrometry (AMS, Wacker et al., 2010), routine and high-precision radiocarbon dating has been extensively applied for the analysis of radiocarbon abundance in riverine POC during the last two decades. Together with the stable isotope composition of carbon ($^{13}C/^{12}C$ ratio, expressed as $\delta^{13}C$), POC content, or other organic-inorganic proxies (*e.g.*, organic carbon-to-nitrogen $C_{org}/N$ ratio, aluminum-to-organic carbon Al/OC ratio), radiocarbon activity helps to constrain the source, transport, and fate of riverine POC (Raymond and Bauer, 2001).

Globally, rivers drain areas of contrasted lithology, climate, tectonics, vegetation, and anthropogenic pressure, parameters that can all impact riverine POC fluxes. At the global scale, riverine $POC_{bio}$ is known to be dominantly sourced from soil organic carbon (SOC) (*e.g.*, Tao et al., 2015; Wu et al., 2018; Wild et al., 2019), whose turnover time (the ratio of OC stock to OC input flux in soil) and thus radiocarbon activity, are greatly controlled by temperature and precipitation (Shi et al., 2020; Eglinton et al., 2021). In permafrost regions, SOC has a longer turnover time and is depleted in $^{14}C$, whereas SOC with the shortest turnover time and the most enriched $^{14}C$ signature is found in tropical forests and savannahs (Shi et al., 2020; Carvalhais et al., 2014). Consequently, riverine POC is significantly older in Arctic rivers (*e.g.*, Kolyma, Lena) than in tropical rivers such as the Congo or Amazon (Holmes et al., 2022; Marwick et al., 2015; Mayorga et al., 2005) due to a major input of aged biospheric OC from thawing permafrost (Wild et al., 2019; Hilton et al., 2015). The geodynamic setting of a river system also exerts a strong control on POC dynamics. In passive margins, terrigenous sediment typically experiences a series of erosion-deposition episodes because of the long distances between the upland source region and the ocean (Blair and Aller, 2012). Consequently, it is on active margins that the original POC source signature is transmitted with the greater fidelity (Blair and Aller, 2012). Finally, humans greatly modify the delivery of fluvial sediment and associated POC to the ocean. In the last decade, sediment delivery in fluvial systems has increased by 215% whereas the net export of riverine sediment to the ocean simultaneously decreased by 49% (Syvitski et al., 2022), indicating the changing amount of eroded POC mobilized to fluvial systems and the final exported POC mass to the ocean  (Stallard, 1998). While land-use change (e.g., soil erosion by

agricultural practices) can lead to increasing terrestrial POC input (Syvitski et al., 2022; Dethier et al., 2022; Montgomery, 2007; Quinton et al, 2010), massive sequestration of POC upstream of dams significantly alters the nature and flux of downstream POC (Syvistski et al., 2022; Maavara et al., 2017; Best et al., 2019; Battin et al., 2009).

Even though recent research has advanced our understanding on the governing environmental factors from catchment- to global scales (Galy et al., 2015; Hilton, 2008; Coppola et al., 2018; Hemingway et al., 2019; Eglinton et al., 2021), there is still a lack of quantitative constraints on the effect of environmental drivers on the carbon isotopic composition of riverine POC. The recent release of the International Soil Radiocarbon Database (ISRaD) (Lawrence et al., 2020) enables to improve Earth system models aiming to predict global SOC radiocarbon distribution and turnover time (Shi et al., 2020; Carvalhais et al., 2014). However, such prediction is still hampered for fluvial POC, despite existing capabilities for modeling water discharge, SPM concentration, and POC content based on global water quality datasets (Ittekkot, 1988; Ludwig et al., 1996, Meybeck, 1993), such as the WBMsed global hydrology model (Cohen et al., 2014) or the Global NEWS2 (Mayorga et al., 2010). Recently, owing to the improved water quality datasets, other sophisticated river biogeochemistry models have been built to understand riverine carbon cycling and environmental inturbations, such as the regional process-based Dynamic In-Stream Chemistry module (DISC-CARBON), but still focus on different carbon fluxes (van Hoek et al., 2021).

Here we provide a new database for riverine POC, called MOREPOC (for MOdern River archivEs of Particulate Organic Carbon) v1.1, compiling 2,283 $\Delta^{14}$C data, thereby representing a significant update of the previously reported global dataset by Marwick et al. (2015) with 531 reported $\Delta^{14}$C measurements . MOREPOC v1.1, featuring data published in international, peer-reviewed articles, provides the basis to 1) uncover the fundamental mechanisms of preservation and alteration of river POC (in terms of "bulk" POC as well as for the individual $POC_{bio}$ and $POC_{petro}$ pools); and 2) help with the construction of numerical models able to simulate the isotopic compositions of POC in the context of global change. MOREPOC database is publicly available on the Zenodo repository at https://doi.org/10.5281/zenodo.7055970 (Ke et al., 2022).

## 2. MOREPOC v1.1: a compilation of data on global riverine POC

### 2.1 Data source

In MOREPOC v1.1, through a comprehensive literature investigation of 115 peer-reviewed articles, we compiled 3,546 POC-related data entries (each entry represents an individual sample), including 2,195 with SPM concentration, 3,053 with POC content, 3,402 with stable carbon isotope $\delta^{13}$C values, 2,283 with radiocarbon activity $\Delta^{14}$C values, 1,937 with total nitrogen content (see details in Table.1), and 299 with aluminum to silicon mass ratios (Al/Si). In addition, reported analytical uncertainties for POC content, $\delta^{13}$C, and $\Delta^{14}$C are included in MOREPOC. Note that riverbed or bank sediments are not included in this database. We selected studies reporting at least one carbon isotopic data, and those with paired elemental and dual carbon isotopic values. Studies reporting only POC contents were not compiled into the MOREPOC v1.1. Potential mistake generated during the compilation of data entries was carefully checked, and duplicate data were removed. A supplementary table "MOREPOC_RM" is provided to give additional information on references, sampling method of SPM,

filtration strategy, carbonate removal methods, and detailed information for the types of acid used to remove carbonate, etc.

**Table 1: Riverine SPM data availability for each continent.**

| Continent | Samples no. | SPM no. | POC no. | TN no. | $\delta^{13}C$ no. | $\Delta^{14}C$ no. |
|---|---|---|---|---|---|---|
| Asia | 1,954 | 1,159 | 1,849 | 1,166 | 1,897 | 1,361 |
| Africa | 291 | 290 | 290 | 103 | 291 | 115 |
| Europe | 130 | 81 | 99 | 23 | 125 | 113 |
| Oceania | 91 | 59 | 91 | 89 | 91 | 26 |
| North America | 793 | 365 | 460 | 411 | 712 | 558 |
| South America | 287 | 241 | 264 | 135 | 286 | 110 |
| Total | 3,546 | 2,195 | 3,053 | 1,937 | 3,402 | 2,283 |

## 2.2 Georeferencing

Location of samples was digitalized if available, and an associate ArcGIS data layer in shapefile format (see MOREPOC_v1.1.rar) is provided with all points projected in a Geographic Coordinate System using the World Geodetic System 1984 (WGS1984). For references only providing a sampling map without any numerical information on sampling location, sampling coordinates were manually extracted using ArcGIS 10.3 after georeferenced adjustment. In the end, 3,339 SPM samples have coordinate information among the 3,546 compiled SPM entries (Figure 1). Furthermore, it can be noted

that most studies chose sampling locations where SPM can be taken as representative of biogeochemical processes at catchment scale, *i.e.*, the river mouth, to better understand the compositions, transport behavior, and fluxes of POC either going to a confluence or an estuary (*e.g.*, Bouchez et al., 2014; Hilton et al., 2015; Holmes et al., 2022).

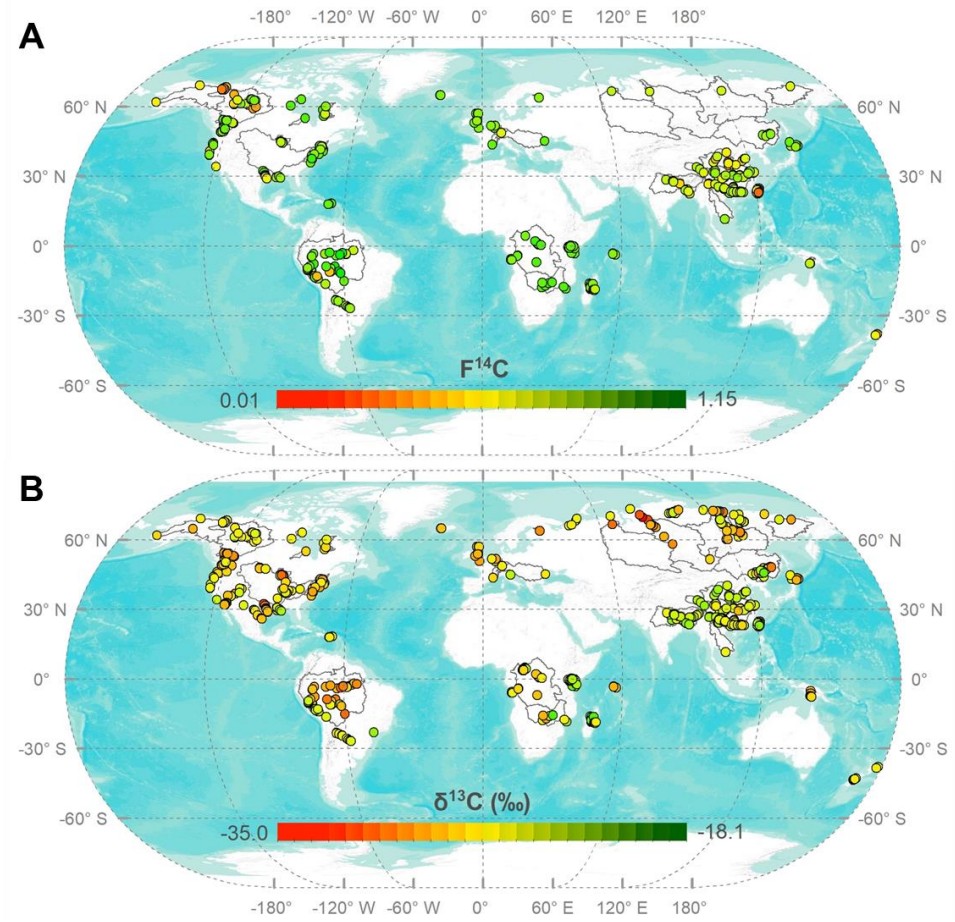

**Figure 1: Overview of the dual carbon isotope data of MOREPOC v1.1. A; Fm$^{14}$C values; B :δ$^{13}$C values. Note that an average value is presented when several samples have been collected at the same location.**

## 2.3 Database structure

To make MOREPOC v1.1 machine-readable, the compiled parameters were labeled with a short name as shown in Table. 2.

**Table. 2 Description of the parameters of the MOREPOC v1.1 database.**

| Parameter | Description | MOREPOC column name |
|---|---|---|
| River name | Name of the major river basin | bas_id |
| Sub river name | Name of the sampled river/stream | riv_id |
| Country | Name of country or places | country |
| Continent | Name of the continent | cont |
| Sampling site/code | Expedition sampling ID | code |
| Sampling date | Time (month/day/year) when the SPM sample was collected | time_m/d/y |
| Latitude | Decimal latitude using WGS 1984 | lat |
| Longitude | Decimal longitude using WGS 1984 | lon |
| Sampling technique | Method of SPM sampling | type_spm |
| Size fraction of SPM | Reported size fractions analyzed | fra_spm |
| SPM concentration (mg/L) | The total dry weight of SPM in mg per liter water column | conc_spm |
| POC concentration (mg/L) | The total dry weight of POC in mg per liter water column | conc_poc |
| POC content (%) | The total POC content of SPM in wt % | perc_poc |
| POC content uncertainty (1$\sigma$) | The analytical uncertainty for POC content (1$\sigma$) | perc_poc_1sd |
| $\delta^{13}C$ (‰) | $\delta^{13}C$ values of POC (carbonate removed) in ‰ | d13C_poc |
| $\delta^{13}C$ uncertainty (1$\sigma$) | The analytical uncertainty for $\delta^{13}C$ of POC | d13C_1sd |
| $\Delta^{14}C$ (‰) | $\Delta^{14}C$ values of POC (carbonate removed) in ‰ | D14C_poc |
| $\Delta^{14}C$ uncertainty (1$\sigma$) | The analytical uncertainty for $\Delta^{14}C$ of POC | D14C_1sd |
| Fraction modern (Fm) | Fraction modern of POC | F14C |
| Radiocarbon ages (year) | Radiocarbon ages before present (1950) | age_14C |
| TN content (%) | The total nitrogen content of SPM in wt % | perc_tn |
| $C_{org}$/N mass ratio | The mass ratio of POC to TN in SPM | cn_ratio |
| Al/Si mass ratio | The mass ratio of Al to Si in SPM | alsi_ratio |
| Reference | Full list of citations of the data source | ref |
| Complete reference | Complete information for cited references | ref_c |
| Measured parameters | Summarization of elemental and isotopic carbon parameters measured | para_m |
| Calculated parameters | Summarization of elemental and isotopic carbon parameters calculated | para_c |
| Filter | Filter used to obtain SPM | filter |
| Acid | The acid type used to remove carbonate in SPM | acid |
| Carbonate removal method | The method used to remove carbonate in SPM | m_acid |

| Acid concentration | The concentration of adopted acid to remove carbonate in SPM | conc_acid |
| Carbonate removal temperature | The environmental temperature for acid to remove carbonate in SPM | temp_acid |
| Carbonate removal duration | The reaction time used for acid to remove carbonate in SPM | time_acid |
| Note | Additional information for carbonate removal process | note |

## 2.4 Information on sampling technique

In the compiled studies, five different sampling techniques (parameter "type_spm" of MOREPOC v1.1) have been adopted to retrieve river sediments with the aim of measuring POC content and composition:

- *Surface SPM sampling ("type_spm = SS")* consists in collecting SPM within the first meter below the channel surface. This sampling scheme is the most frequently used and widely adopted in riverine POC studies.

- *Mid-depth SPM sampling ("MS")* consists in collecting SPM at an intermediate depth between the river surface and bottom. This sampling strategy has been used in studies on the Mekong (Martin et al., 2013) and the Mackenzie (Campeau et al., 2020).

- *Integrated sampling over depth profiles ("ISD")* aims at obtaining a representative SPM sample accounting for grain size sorting along the water column, typically by making a flux-weighted composite of several samples collected at different depths along the water column. This sampling strategy has been adopted only for the Huanghe and the Changjiang (Wang et al., 2012), and for the Zengjiang, a tributary of the Zhujiang (Gao et al., 2007).

- *Point sampling along depth profiles ("PSD")* is the collection of SPM along individual depth profiles at different depths in the water columns. In this method and in contrast to the previous one, each SPM sample is treated and analyzed separately. This method allows accessing the full range of particle sizes of SPM, explaining its wide use in the literature (*e.g.*, Ganges-Brahmaputra [Galy et al., 2008a, b], Mackenzie [Hilton et al., 2015], Bermejo [Repasch et al., 2021]).

- *Point sampling over transects ("PST")* corresponds to *PSD* collection of SPM along several depth profiles across a given river channel section. This sophisticated sampling scheme allows for the exploration of the potential lateral heterogeneity in a river channel. It has been recently used in the Amazon (Bouchez et al., 2014), the Salween and the Irrawaddy (Baronas et al., 2020), and the Danube (Freymond et al., 2018).

## 2.5 Information on SPM extraction from river water samples and on analysed size fractions

Broadly speaking, two methods are commonly adopted to extract SPM from a water sample, 1) continuous flow centrifugation, whereby large volumes of water can be centrifuged at high centrifugal forces; 2) filtration under pressure or vacuum using membranes made of glass fiber (GF/F), PolyEtherSulfone (PES), Polycarbonate (PTCE), Nylon, quartz fiber, or Mixed Cellulose Esters (MCE), at a mesh size ranging from 0.2 μm to 1.0 μm. This information is recorded in MOREPOC_RM in detail if described in the corresponding source reference (parameter "Filters").

In general, most studies used bulk SPM retrieved after filtration for the analysis of POC. However, in some studies, only

certain size fractions of SPM were analyzed, after separation into *e.g.,* a fine (<63 μm) and a coarse (>63 μm) fraction. This information is reported in MOREPOC v1.1 as the "fra_spm" parameter (see Table 2).

## 2.6 Information on carbonate removal method

Particulate inorganic carbon (PIC) and POC have distinct carbon isotopic signatures, such that the accuracy of POC $\delta^{13}C$ and $\Delta^{14}C$ values could be compromised if the PIC is not efficiently removed by acidification prior to POC analysis (Komada et al., 2008). Three methods have been adopted in the studies referenced in MOREPOC v1.1:

- The *"acid rinse method"*, in which sediment samples are soaked with diluted acid at a given temperature for a given time, and then rinsed with distilled water.
- The "*acid vapor method*", in which sediment samples are exposed to vaporous concentrated hydrochloric acid in a closed system maintained at a given temperature for a given time, and then evacuated under vacuum.
- The "*acid infiltration method*", in which sediment samples are infiltrated in-situ in silver capsules with diluted hydrochloric acid, and then subjected to drying.

The "acid rinse method" and "acid vapor method" have been widely used by the community to remove carbonates from sediments, the "acid infiltration method" is also a common carbonate-removal method but is only reported by Menges et al., 2020 in MOREPOC v1.1.

In addition, a separate file of MOREPOC v1.1 ("MOREPOC_RM") provides detail on carbonate removal method (*m_acid*), acid type (*acid*), molarity and quality (*conc_acid*), reaction time in unit of hours (*time_acid*), and reaction temperature in Celsius degrees (*temp_acid*), allowing for quality evaluation of the method used in the cited references.

## 2.7 Definitions of POC composition variables and units

In MOREPOC v1.1, all data are either taken directly from references or calculated from the reference data. POC content (POC%), and total nitrogen content (TN%) are reported as dry weight percentage (%). Besides, POC concentration in river water (mg/L) can be calculated using SPM concentration reported as dry weight per liter (mg/L) and percentage content of POC (%).,

Most importantly, the fundamental component of MOREPOC v1.1 consists of an extensive dataset for stable carbon isotope values ($\delta^{13}C$, in ‰ relative to VPDB) and radiocarbon compositions (provided as both $\Delta^{14}C$ in ‰ or as $F^{14}C$; see below). Fraction modern, $F^{14}C$, is the deviation of a sample's 14C atoms from that of the modern standard. Conventional Radiocarbon Ages (RCA) are given in MOREPOC v1.1 following Stuiver and Polach (1977), using the Libby half-life of 5,567 years with the mean life of 8,033 for $^{14}C$. RCA is expressed in units of years before present (BP), with year zero being 1950:

$$RCA = -8033 \ln (F^{14}C) \tag{1}$$

$\Delta^{14}C$ value, which is defined as the relative difference between the absolute international standard (the base year 1950) and sample activity corrected for age and mass-dependent fractionation (Stuiver and Polach, 1977), is reported in MOREPOC v1.1

as well. A positive $\Delta^{14}C$ indicates the presence of "bomb carbon", whereas a negative $\Delta^{14}C$ indicates that the radioactive decay of C overwhelms any incorporation of bomb carbon into the sample. The $\Delta^{14}C$ calculation is defined as equation 2:

$$\Delta^{14}C \text{ (in ‰)} = \left[ F^{14}C * exp\left( \frac{1950-yr}{8267} \right) - 1 \right] * 1000 \tag{2}$$

Where *yr* is the year when the sediment was collected, 8,267 is the true mean life of $^{14}C$ using the Cambridge half-life of 5,730 years.

The term fraction of modern ($F^{14}C$) is adopted in the above equations, $F^{14}C$ is defined as equation 3 (Donahue et al., 1990):

$$F^{14}C = \frac{\left( \frac{^{14}C}{^{13}C} \right)_{sample[-25]}}{0.95\left( \frac{^{14}C}{^{13}C} \right)_{OxI[-19]}} \tag{3}$$

Where the denominator is 95% of the $^{14}C$ activity of the Oxalic Acid I (OxI) standard material in 1950, and the numerator is corrected for fractionation to a common $\delta^{13}C$ value of -25‰.

Lastly, if available, the aluminum-to-silicon mass ratio (Al/Si) is also provided in MOREPOC v1.1. This elemental ratio is an efficient proxy for the particle size of riverine sediment, allowing to characterize the grain size effect of sediments on POC loading in fluvial delivery (Galy et al., 2008b; Bouchez et al., 2011; Hilton et al., 2015). The mineralogy and particle size of sediments are generally related, with coarse particles being quartz-rich (low Al/Si ratios) and fine particles being clay-rich (high Al/Si ratios) (Galy et al., 2008b). POC contents are usually positively correlated with proportions of fine-grained fractions (Mayer, 1994; Galy et al., 2008b; Bouchez et al., 2014).

## 2.8 Extent of MOREPOC v1.1

Although MOREPOC v1.1 features data from river systems worldwide, it does not offer the same degree of representativeness for all the continents with, for instance, an over-representation of Asian rivers and an under-representation of rivers draining Europe and Oceania (Table. 1). It can be noticed that there are relatively abundant POC studies in North America fluvial systems. However, MOREPOC database also indicates the lack of studies on POC in fluvial systems in the cryosphere regions such as Antarctica and Greenland as well as in arid regions, including Australia, and vast areas spanning from northern Africa to middle east Asia (Figure 1).

MOREPOC v1.1 database does not compile elemental and dual isotopic compositions of molecular compounds (plant-wax fatty-acid and lignin-phenol), thermal labile fractions, or black carbon. However, such complementary data could be incorporated into future versions.

## 3. Global riverine POC patterns

### 3.1 Trends of $\delta^{13}C$ and $\Delta^{14}C$ in MOREPOC v1.1

The mobilization of terrestrial organic matter into fluvial systems depends on the interplay between tectonics, climate,

geomorphology, lithology, and anthropogenic activities, all controlling to some extent the amount and composition of riverine POC (Blair and Leithold, 2010; Eglinton et al., 2021).

Riverine POC displays significant heterogeneity in elemental and isotopic compositions of carbon around the globe (Figures 1 and 2). $\delta^{13}C$ values (n=3,402) range from -38‰ to -17‰ with an average value of -26.3‰. As shown in Figure 2, the majority of the data falls between -28‰ and -24‰ (n=1,770, 52.0% of total entries), which is consistent with the overall isotopic signature of the terrestrial biosphere of -26±7‰ (Schidlowski et al., 1988). The age of riverine POC spans from "modern" (that is, recording bomb-derived carbon) to "ancient" (strongly influenced by fossil petrogenic source), with $\Delta^{14}C$ values (n=2,283) ranging from -990.1‰ to 147.7‰ with a statistical average of -386‰. A large fraction of the $\Delta^{14}C$ values (n=775, 33.9% of total entries) falls within the range -300‰ to 0‰, this range dominates the database in Marwick et al (2015) as well (n=278, 52.3% of Marwick's total entries). The MOREPOC v1.1 dataset is on average more $^{14}C$ depleted than that of Marwick et al (2015).

Around the globe, the most ancient POC ($\Delta^{14}C$ = -990‰) is found in small mountainous rivers in Taiwan (Hilton et al., 2010), in which the entirety of POC is derived from the erosion of sedimentary rocks. In the riverine POC dataset of MOREPOC v1.1, bomb carbon signals are abundant ($\Delta^{14}C > 0$‰), particularly for African rivers in tropical regions such as Athi-Galana-Sabaki, Tana, Zambezi, and Congo (Marwick et al., 2015; Spencer et al., 2012); rivers in North America including Hudson, Siuslaw, and York; rivers draining to the Hudson Bay (Leithold et al., 2006; Raymond and Bauer, 2001; Godin et al., 2017; Longworth et al., 2007); and the Andean Amazon (Mayorga et al., 2005; Townsend-Small et al., 2007). Around the Qinghai-Tibet Plateau, where most large river systems in eastern and southern Asia share similar high elevation source regions, POC is usually characterized by relatively depleted $^{14}C$ signals due to high erosion rates of sedimentary rocks in mountainous regions, like in the Ganges-Brahmaputra (Galy et al., 2007) or the Changjiang (Wang et al., 2012; Wang et al., 2019), etc., and erosion of soils containing pre-aged OC, *e.g.* Huanghe (Tao et al., 2015). The most depleted $^{13}C$ signatures (less than -35‰) are observed for POC from Arctic rivers, such as the Ob', Yukon, and Kolyma (Holmes et al., 2022). The highest $\delta^{13}C$ values (higher than -22‰) are found in rivers in Africa (Athi-Galana-Sabaki, Betsiboka, and Tana; Marwick et al., 2015; Tamooh et al., 2013) and mountainous rivers (*e.g.*, Taiwan [Hilton et al., 2010], upper Ganges [Galy et al., 2007, 2008b], Minjiang [Wang et al., 2019], etc.)).

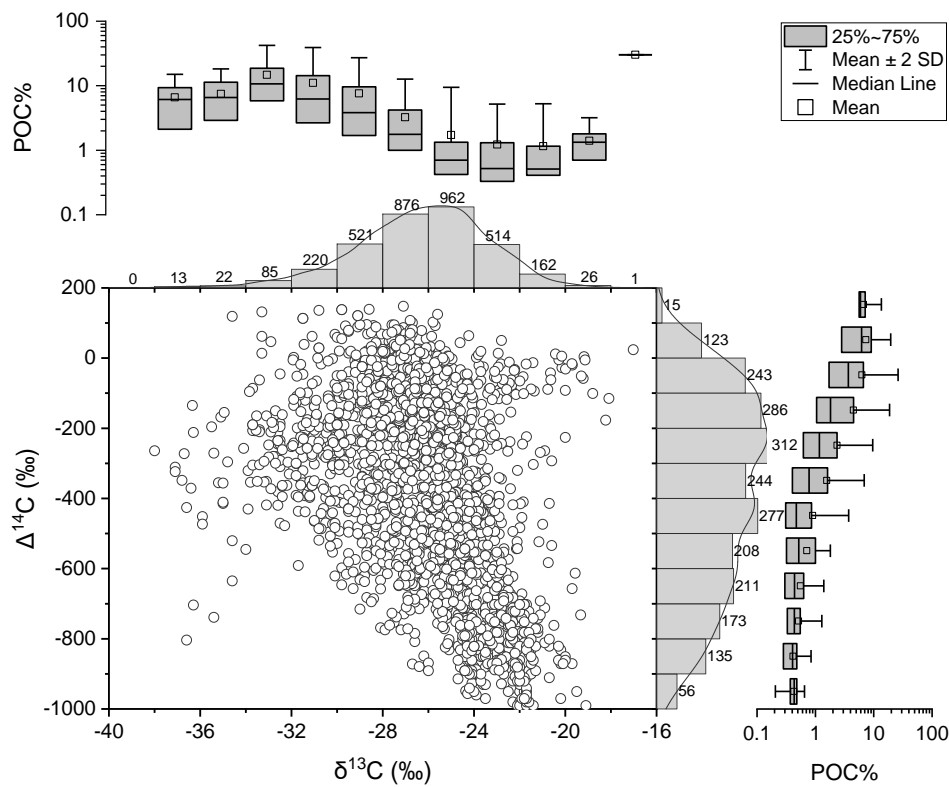

**Figure 2: δ¹³C versus Δ¹⁴C of MOREPOC v1.1 (n=2,129). Frequency distribution histograms for δ¹³C (x-axis) and Δ¹⁴C (y-axis) are shown, with δ¹³C values binned every -2.5‰ from -40‰ to -15‰, and Δ¹⁴C values binned every -50‰ from -1000‰ to 200‰. Each bin is labeled with the number of samples it hosts. Solid lines represent the corresponding probability density functions. Box charts represent the statistical analysis of POC% in each bine of δ¹³C (x-axis) and Δ¹⁴C (y-axis).**

As observed in the global compilation in Figure 2, elemental and isotopic data of POC generally show an inverse relationship between δ¹³C and Δ¹⁴C, and generally an increasing POC content with higher radiocarbon activity of POC. Indeed, OC from sedimentary rocks (*i.e.,* dead OC with Δ¹⁴C=-1000‰ by definition) usually has ¹³C-enriched signatures compared to recent biomass. Eroded material from sedimentary rocks thus has lower POC content, ¹⁴C-depleted signatures and relatively high δ¹³C signatures. This global pattern stems from the global dominance of C3 plants in the compiled catchments (Figure 2). However, POC-rich riverine SPM can also be relatively enriched in ¹³C, *i.e.*, δ¹³C values larger than -20‰ (Figure 2 and Figure 3). This pattern indicates the presence of an additional pool of ¹⁴C- and ¹³C-rich POC in the terrestrial environment (Cerling et al., 1997), consisting of modern C4-plants in catchments dominated by grasslands or savannah (*e.g.,* Marwick et al., 2015). The maximum values of δ¹³C and Δ¹⁴C of POC (dotted line in Figure 3) tend to be more depleted at high latitudes than at low latitudes. This might reflect the major POC components: 1) dominated by $POC_{bio}$, the combined effects of increasing coverage of C4 plants in tropical regions and the input of pre-aged $OC_{bio}$ of C3 plants from degrading permafrost at high latitude (Cerling et al., 1997; Still et al., 2003); 2) dominated by $POC_{petro}$, rivers in mountainous regions tends to erode ¹³C-enrich petrogenic OC (Hilton et al., 2010; Galy et al., 2007). In addition, aquatic authigenic production can be an important mechanism

contributing [13]C-depleted and [14]C-enriched POC (Longworth et al., 2007; Marwick et al., 2015; Wu et al., 2018).

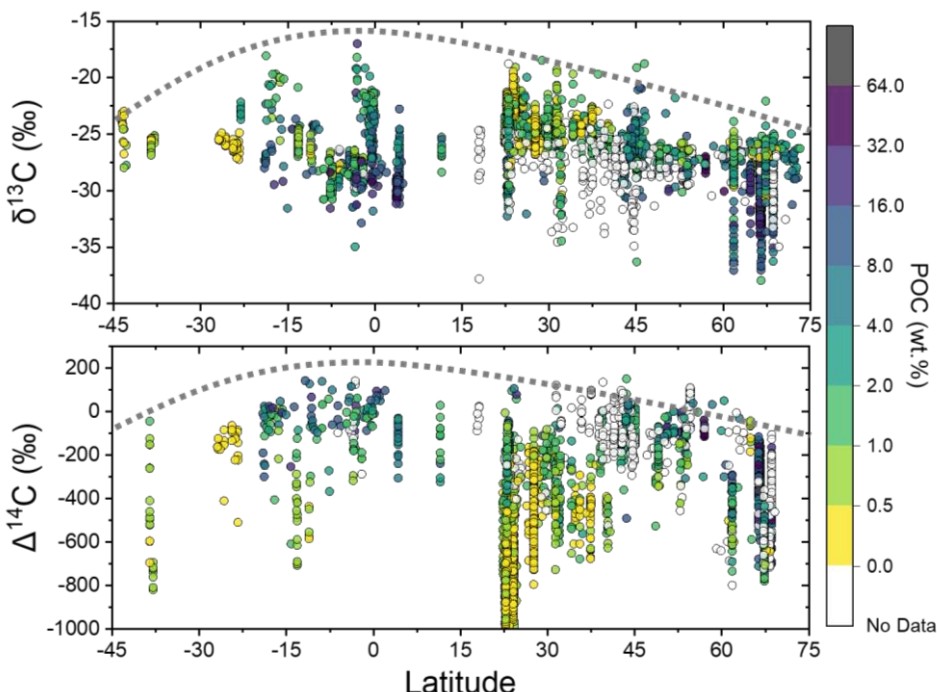

**Figure 3: Latitudinal trends in δ[13]C (n=3,204) and Δ[14]C (n=2,212) in MOREPOC v1.1. Colors indicate POC content (wt. %). Dotted lines represent upper envelopes of δ[13]C and Δ[14]C values of POC.**

### 3.2 Relationships between riverine SPM and POC

MOREPOC v1.1 features data from rivers with SPM concentrations ranging from 0.35 to 199,000 mg/L with POC content from 0.01% to 91.67%. SPM and POC concentrations (both expressed in mg/L; n=2,115) are positively correlated (Figure 4). However, the global trend shows that an increase in SPM concentration is accompanied by a decrease in POC content (in %), which is largely owing to a dilution effect by inorganic materials (Figure 4, Ittekkot, 1998; Ludwig et al., 1996; Meybeck, 1993). In MOREPOC v1.1, large SPM concentrations (over 10,000 mg/L) are generally observed in mountainous rivers, such as the Choshui and Liwu rivers in Taiwan (Hilton et al., 2008; Kao et al., 2014), the Santa Clara River (USA) (Masiello and Druffel, 2001), or the Minjiang (a major tributary of the upper Changjiang, China) (Wang et al., 2019). The Huanghe is an exception in that it has very large SPM concentrations in its middle reaches where it drains the Chinese Loess Plateau (Qu et al., 2020; Hu et al., 2015). Although the sediment of highly turbid rivers is typically POC-poor, high sediment concentrations generate the largest POC export rates (Figure 4). This observation also underlines the importance of sediment transport near the channel bottom in large rivers where SPM concentration is usually much higher than at the surface (Figure 5, e.g., Ganges-Brahmaputra-Meghna [Galy et al 2007, 2008b], Mackenzie [Hilton et al., 2015], Amazon [Bouchez et al., 2014], and Yukon (Holmes et al., 2022) etc.), as well as the role of stochastic events leading to high-turbidity episodes such as storms, landslides,

or earthquakes (Hilton et al., 2008; Wang et al., 2015; Frith et al., 2018). Small SPM concentrations (less than 10 mg/L) are generally found in rivers during the frozen season or rivers draining either high-latitude or tropical areas characterized by low-relief settings, in which POC content is relatively high (Gao et al., 2007; Holmes et al., 2022).

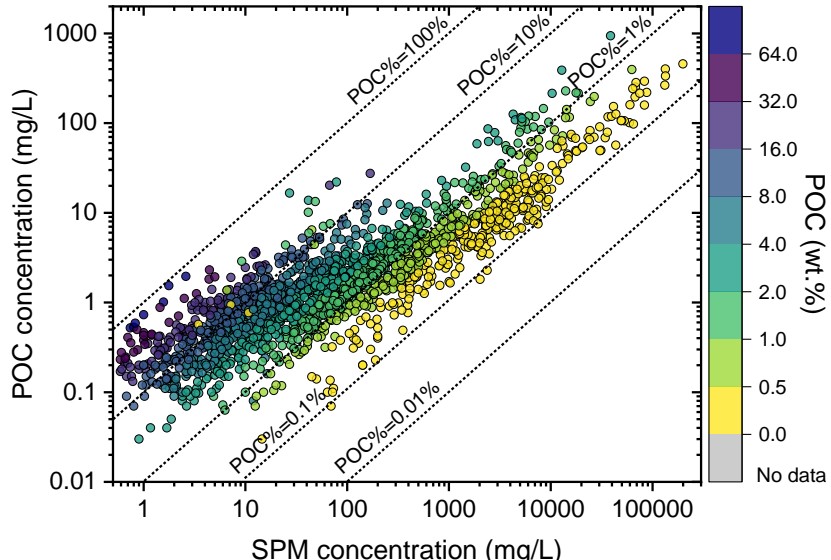

**Figure 4: River SPM concentration vs. POC concentration (n=2,135), both expressed in mg/L. Dotted lines represent contours of constant POC content. Colors indicate the POC content from data entries in MOREPOC v1.1.**

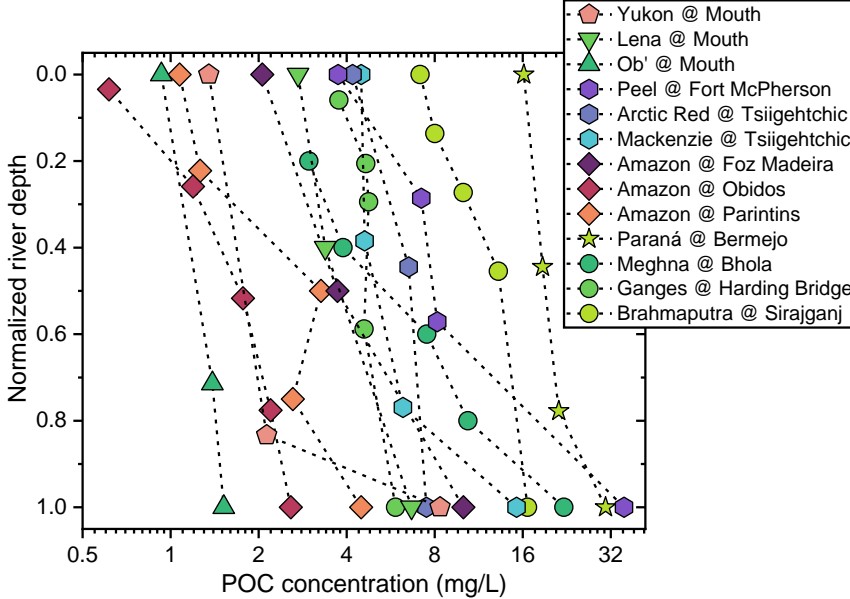

**Figure 5: POC concentration variation in vertical water columns from depth profiles in global large rivers. Selected depth profiles are from the Yukon, Lena, and Ob' (Holmes et al., 2022), Mackenzie including Peel and Arctic Red (Hilton et al., 2015), Amazon (Bouchez et al., 2014), Paraná (Repasch et al., 2021), and Ganges-Brahmaputra-Meghna systems (Galy et al., 2007, 2008b). Normalized river depth is calculated by normalizing the individual sample depth to the maximum sample depth of the corresponding profile.**

In general, POC becomes $^{14}C$-depleted with increasing suspended sediment load and decreasing POC content. These patterns are most likely caused by the dilution of $POC_{bio}$ by $POC_{petro}$ in areas of strong erosion (Leithold et al., 2016). However, MOREPOC v1.1 also highlights that low SPM load associated with high POC content is often characterized by significantly low $\Delta^{14}C$ values (Figure 6, 7). Most of those samples come from Arctic river systems. This rises some concerns because Arctic permafrost soils store approximately twice the current amount of carbon contained in the atmosphere (Zimov et al., 2006), and biospheric OC that was previously stored in frozen soils over thousands of years is being released and can induce accelerated environmental changes (Schuur et al., 2015; Vonk et al., 2015; Wild et al., 2019). How OC in permafrost regions responds to global warming should be a key research issue in future studies. Meanwhile, it is worth noting, for a given SPM concentration, $^{14}C$ abundance can be contrasted. For example, rivers draining low-latitude, tropical regions (especially 10°N – 10°S; *e.g.,* African rivers) or high-latitude regions (60°N - 75°N; *e.g.,* Siberian Arctic rivers) are usually characterized by relatively low SPM concentration and abundant POC composition. Nevertheless, riverine POC from the low-latitude African rivers is much younger compared to that from the Arctic Siberian regions. This difference most likely stems from the contrasting radiocarbon activities and turnover time of the soil organic carbon between these two regions, which are primarily driven by climate (Eglinton et al., 2021; Marwick et al., 2015; Wild et al., 2019; Vonk et al., 2015).

The MOREPOC v1.1 dataset also reveals that under a given climate, river systems can be heterogeneous in terms of SPM concentration and associated POC composition. For example, amongst circum-Arctic rivers, the Mackenzie River has a relatively large SPM concentration of 135.1 mg/L on average (1σ=16.6, n=106) with 3.01% POC (1σ=0.39, n=105) characterized by a fairly low $\Delta^{14}C$ (average value of -599.5‰, 1σ=7.7, n=118) near the river mouth (Hilton et al., 2015; Schwab et al., 2020; Holmes et al., 2022). In contrast, the Yenisei River only has an average SPM concentration of 5.2 mg/L (1σ=0.6, n=86) but much higher POC contents (18.0%, 1σ=2.1, n=83) and $\Delta^{14}C$ values (-342.3‰, 1σ=15.7, n=66; Holmes et al., 2022). Such difference suggests that lithology and geomorphology can play an important role in riverine POC composition and load by providing a substantial fraction of fossil OC (Hilton et al., 2015). On the other hand, small mountainous rivers such as those in Taiwan or those draining the Himalayas show large SPM concentrations and low POC contents with low radiocarbon activities. These regions characterized by active tectonics, steep slopes, and intense precipitation, act as global hotspots for sediment production and thus petrogenic OC mobilization (Milliman and Farnsworth, 2011; Hilton and West, 2020).

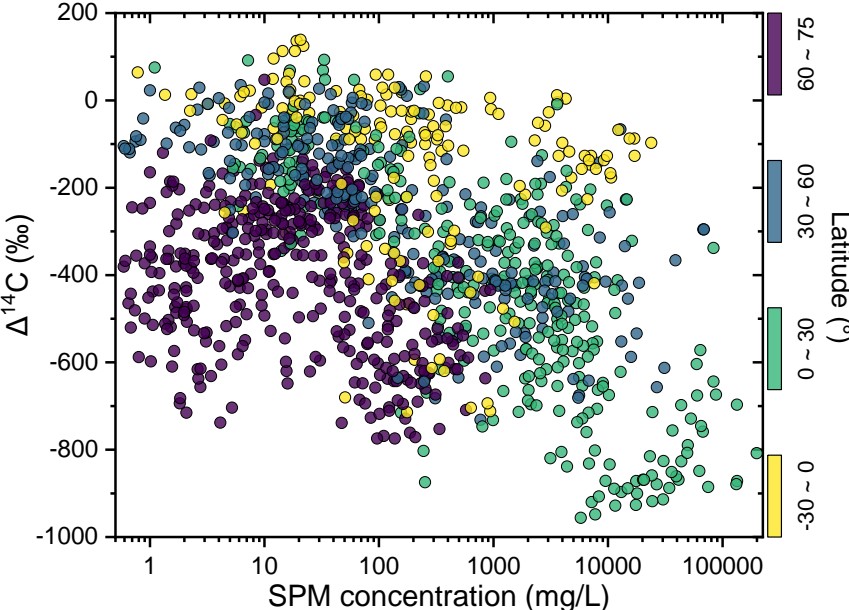

**Figure 6: POC Δ¹⁴C vs. SPM concentration (n=1157). Colors indicate the latitude of the sampling location. Note the log-scale used for SPM concentration.**

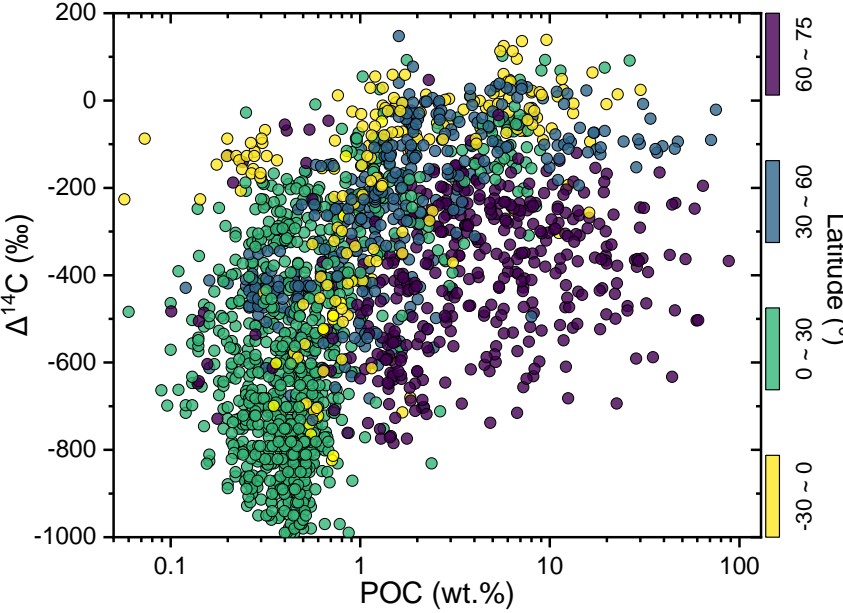

**Figure 7: Δ¹⁴C values vs. POC content (n=1,860). Colors indicate the latitude of the sampling location. Note the log scale used for POC content.**

## 4. Database availability

MOREPOC v1.1 database is publicly available on the Zenodo repository in machine-readable formats as Excel spreadsheet (.xslx), comma limited table (.csv), and GIS shapefile at https://doi.org/10.5281/zenodo.7055970 (Ke et al., 2022).

## 5. Conclusions

In this paper, we introduce MOREPOC, the largest and most comprehensive database for riverine suspended particulate matter (SPM) concentration and particulate organic carbon (POC) composition, including POC and total nitrogen (TN) content, stable carbon isotope ($^{13}$C), cosmogenic-radioactive carbon isotope ($^{14}$C), as well as aluminum-to-silicon (Al/Si) mass ratios. MOREPOC will benefit the scientific community carrying out research on riverine POC sources, transport, and fate. Furthermore, it will help feed and validate Earth system models in order to improve the ability of models to constrain all the components of the global carbon cycle. Combined with ocean sediment databases, such as CASCADE (Circum-Arctic Sediment Carbon DatabasE, Martens et al., 2021) or MOSAIC (Modern Ocean Sediment Archive and Inventory of Carbon, Van der Voort et al., 2021), MOREPOC will enable a better understanding of the fate of POC from the terrestrial source to sink at the ocean bottom. Existing environmental raster global datasets for climate, geomorphology, lithology, tectonics, hydrology, and land use, also offer promising prospects for the use of MOREPOC for identifying the controls on POC fluxes and composition, in particular using advanced statistical analysis or machine learning techniques. Future updates of MOREPOC should include new bulk POC parameters as well as data on molecular fractions, thermal labile fractions, or specific components such as black carbon or fossil carbon, which should, in turn, provide additional insight into the alteration of riverine POC from source to sink, an essential feature of the global carbon cycle.

## Author contribution

YK collected the MOREPOC data and conceptualized, designed, structured, and filled the database. YK and DC contributed to the database checking. YK prepared the manuscript. YK drafted and coordinated the manuscript with input from DC, JB, and CQ.

## Competing interests

The authors declare that they have no conflict of interest.

## Acknowledgments

The development of MOREPOC database was funded by the Agence Nationale de la Recherche (ANR) SEDIMAN (Grant

ANR-15-CE01-0012), the authors acknowledge the Ph.D. scholarship awarded to Yutian Ke (No. 201706180008) by the China Scholarship Council.

**Financial support**

This research has been supported by Agence Nationale de la Recherche (ANR) SEDIMAN (Grant ANR-15-CE01-0012) and
the China Scholarship Council (Grant 201706180008).

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
