# Peer review of "MOdern River archivEs of Particulate Organic Carbon: MOREPOC"

_Earth System Science Data, 2022_

## Referee Comment (RC1)

**Review of "MOdern River archivEs of Particulate Organic Carbon: MOREPOC" by Yutian Ke, Damien Calmels, Julien Bouchez, and Cecile Quantin for publication in *Earth System Science Data*.**

This dataset is a comprehensive inventory of TOC, $\delta^{13}$C, F$^{14}$C, C:N ratios, Al:Si ratios, and important methods related metadata for particulate organic carbon collected and analyzed from rivers around the world. This builds significantly upon previously compiled datasets, which have an order of magnitude fewer datapoints that this new dataset submitted by Ke and colleagues. The publication of this dataset is timely, as the number of studies measuring the geochemistry of fluvial POC has increased over the past decade, and more studies are adopting the dual isotope measurement approach. I am excited about the future studies this dataset will enable.

This dataset seems to be thorough with respect to including all available published data and the dataset includes the relevant parameters and metadata needed to understand how the data were collected. However, there are some issues with the dataset, particularly with respect to reporting measurement uncertainties, naming the variables used to represent the data, and formatting and populating sampling dates. There are also minor issues throughout the text that need to be addressed before this manuscript can be published.

After the dataset and manuscript have been revised to address all issues detailed below, I support publication of this manuscript in *Earth System Science Data*.

**Detailed comments:**

*Dataset:*

One significant issue with this dataset is that there are no uncertainties reported for the POC $\delta^{13}$C, F$^{14}$C, and radiocarbon age measurements. Analytical uncertainties are typically required when publishing these isotopic measurements, so they should be available in most published datasets. Please add these as columns in the dataset.

The names of the first two columns of the dataset (riv_na and bas_na) are not intuitive. Without looking carefully, I would interpret riv_na to be the name of the sampled river, which bas_na would be the name of the major drainage basin that river is in. To ease use of the dataset, I recommend switching these column headers so that riv_na is the name of the sampled river and bas_na is the name of the larger drainage basin to which the sampled river contributes.

With respect to the river name, I suggest removing the country name from the river name and adding a separate column for the country. In its current format, if someone wants to filter the dataset by specific river, they need to type in the country in parentheses after the river name, which makes data analysis challenging.

CN_mar and as_mar also are not intuitive variable names. CN_ratio and alsi_ratio seem more appropriate and do not exceed the maximum number of characters used for other variables in the dataset.

Rca could also be changed to age_14C.

91 samples do not have dates associated with them, which I assume is because the dates were not reported in the publication. I would appreciate if the authors could confirm whether they double checked the data sources for information on sampling dates.

549 of the reported measurements do not have individual dates, but a range of years (e.g., 1999-2004). This is not very useful for someone who wants to calibrate POC data to points on river hydrographs. It is also unclear whether the reported geochemical measurements are from individual samples, or a calculated composite measurement of multiple samples. This either needs to be explained in the text, or the dataset needs to be edited to breakout the samples into individual dates.

One of the samples (Amazon River at Obidos, 2005) has the latitude and longitude reversed, such that it plots down near Antarctica. It appears that this point was removed from Figure 1, but it is still in the spreadsheet and in the shapefile.

***Manuscript text:***

Line 7-8: This sentence could be re-organized to digest more easily. I suggest "Riverine transport of particulate organic carbon (POC) associated with terrigenous solids to the ocean has an important role in the global carbon cycle."

Line 31: "…over geological timescales (>100,000 years)."

Line 45: While SOC in permafrost regions is certainly depleted in 14C because it is old and 14C still decays over time, this organic matter may not have a long turnover time. When frozen, its decomposition rate is zero, but when thawed it may actually have a fast turnover time. We don't yet have enough data to constrain the turnover time because the turnover time "clock" is only set once the permafrost thaws, which has been occurring more recently.

Line 47: The Carvalhais et al. (2014) study shows total ecosystem carbon turnover times, not necessarily soil organic carbon turnover time and is not constrained by 14C data, so using this reference is a misleading.  The Shi et al. (2020) reference is appropriate here.

Line 53: Write "in reservoirs" rather than "at reservoirs."

Line 66: Suggested edit: "…with 531 reported d14C measurements."

Line 81: "Decarbonization" is not an appropriate word to use here. I suggest "carbonate removal methods." It is also unclear what is meant by "acid adopted." Do you mean type of acid used for removing carbonate? If so, please reword for clarity.

Table 2:

- Decarbonization is not an appropriate term to use here, because it implies removing all carbon. Rather, the authors should use "carbonate removal."

Line 126: Again, not sure that decarbonization is the right term to use to describe carbonate removal. It implies that all carbon is removed from the samples. It is more straightforward to say "carbonate removal method" or "inorganic carbon removal method."

Lines 134-136: What are the units associated with time and temperature of acid treatment? This is also no reported in the data spreadsheet.

Line 142: "…consists *of* an extensive dataset…"

Line 146: There should be a comma or colon after 1950, not a quotation mark.

Lines 142-154: Both of these equations have F14C as an input, but the authors never provide an equation to derive F14C from an AMS measurements of $^{14}C/^{12}C$. I recommend adding the equation for $F^{14}C$:

$$F^{14}C = \left[ \frac{\left(\frac{^{14}C}{^{12}C}\right)_{sample,-25}}{0.95\left(\frac{^{14}C}{^{12}C}\right)_{OX1,-19}} \right]$$

Where the denominator is 95% of the 14C activity of the Oxalic Acid 1 standard material in 1950, and the numerator is corrected for fractionation to a common d13C value of -25 per mil.

Line 170: "range" not "ranges"

Line 259: Are Earth system modelers not a part of the scientific community? This sentence needs to be re-arranged or re-worded. I suggest deleting "as well as Earth system modelers." It would be worth adding a sentence mentioning that having such a dataset to work with can help inform and validate Earth system models, improving our ability to model the global carbon cycle.

---

## Referee Comment (RC2)

**Review**

**"MOdern River archivEs of Particulate Organic Carbon: MOREPOC"**

*by Yutian Ke, Damien Calmels, Julien Bouchez, and Cecile Quantin*
*for publication in Earth System Science Data.*

*15.08.2022*

Ke and colleagues provide a large (~ 4x the size previously compiled sets) compilation of published data on riverine particulate organic carbon (POC, incl. isotopic composition and related N content), suspended particulate matter concentrations (SPM),and Al/Si weight ratios of the corresponding sediment. This comprehensible dataset is f good quality and accompanied by a wealth of metadata, such as geographical or methodological information, improving its interpretability and usability of the database. However, uncertainties are commonly reported alongside carbon isotopic values and could be integrated into the database. Clarity and variable naming could also be improved. Otherwise, I have only a few minor comments regarding the database (see below).

The descriptive article adequately summarizes database content, structure and  patterns within the data. It gives most background information necessary to understand relevance, quality and acquisition of the data. At times the article is written too much in a POC-expert language and misses a few explanations necessary to fully understand the data. More specific comments are attached.

This publication seems timely, relevant and useful to the Earth science community and, generally, researchers interested in riverine and coastal organic matter processes and carbon cycling. The size and high spatial coverage of the set provide a proper statistical basis and will certainly help improving our understanding of terrestrial and marine carbon cycling. Detailed comments on data and article can be found below. After these issues have been addressed, I strongly support publication in Earth System Science Data.

**Data Comments**

*General*

- consider adding readme with variable & unit information. Consider providing here also values and equations used.

- some variable names could be more intuitive (riv_na, bas_na)

- Could uncertainties be added, at least of the carbon isotopic values, where uncertainty is usually reported? I acknowledge that this is (unfortunately) not really common practice in geochemical edatabase work, but would improve the interpretability (and thus value) of the database significantly.

*In MOREPOC v1.0_RM*

- add which variables given in which paper (apart from 14C & 13C)?

*In MOREPOC v1.0*

- Origin of values is not always clear: e.g., SPM of Santa Clara in Masiello & Druffel (2001) is derived from USGS monitoring & extrapolation based on relationship of SPM and water discharge. Couldn't find the actual numbers given in the database within the paper by Masiello & Duffel, except for 1 sample (March 25, 1998 with 17,813 mg/L in database and 17.,230 mg/L in M&D paper). How comes there is a difference? Did you use POC conc. and POC wt% to derive SPM conc.? Please indicate whether individual variables are originally given in the corresponding paper and/or more detail on how variables were derived..

- some variables could be formatted more user friendly ()

> - RCA '0' instead of 'modern' too keep it numerical? Also '14C_age' or similar could be a more intuitive name

> - No brackets, points, commas or empty spaces in the data fields, especially in text 'values' (or strings)

> - Separate countries and continent from 'cont' variable.

**Article Comments**

**General:**

The text is well structured and easily readable. It gives a good background and basic understanding to the database published. However, it could be written in less expert-language, could better explain notations/values and citations could be more thorough/original.

**Specific:**

L21: Not only input but also susceptibility to mineralization and specific environment where its deposited are key features of carbon cycling (see Blair & Aller 2012, who are cited just before).

L24: The 'biogenic' source could be split into river-authigenic, land plant (litterfall) and soil organic carbon. Generally, riverine authigenic POC generally seems to be a little underappreciated in the article.

L32: Are you referring global carbon fluxes of biogenic POC sequestration? Clarify and give estimates of these fluxes?

L35: 'played by' could be 'of' to be slightly more concise.

L47-48: Evtl. highlight here the contribution of old (petrogenic) carbon release from thawing permafrost?

L.50-51: It is also because of sedimentary dynamics (e.g., steady accumulation scenario vs. fluid and mobile mud layers) and increased oxygen exposure in highly energetic systems (the 'incinerator' concept of Blair & Aller).

L.54: Add reference: Dethier et al. 2022 Rapid changes to global river suspended sediment flux by humans. Science 376(6600), pp.1447-1452. There are also considerations on the impact of humans on riverine carbon cycling (by Taylor Maavara et al. 2017, Nature, or van Hoek et al. 2022, ES&T, among others). It would be suitable for the paper to address this here.

L61-62: Water quality datasets have significantly improved since 1996. This should be acknowledged here. There are also more recent and increasingly sophisticated model of riverine carbon cycling and fluxes (summarized e.g., in van Hoek et al. 2022, ES&T). These should also be acknowledged here.

L.65: Generally it would help the reader, especially if non-expert in respect to carbon isotopes, if notations (here D14C) and units (section 2.7) could be explained before using them in the text. Fm14C is never explained, not even in section 2.7, but extensively used.

L.79: Which statistical examinations were used?

L155: If space allows it may be instructive to mention Al/Si can also be related to specific surface area to derive carbon loading (mass OC per mineral surface area).

L.158: Also no data from Greenland and arid regions (N Africa & Arabia) as far as I can see from Fig 1.

L.159: If space, add a sentence or two on the other continents?

L.161: Maybe add to this suggestion a note on what could be done with this data and what would be the added value for insights into POC processes?

L.166: There is also control by biology/vegetation and by in-stream and generally fresh-water processes.

L178-179: The Fm14C in SE Asia (Fig 1) should also be mentioned here!

L.197: This explanation seems a bit too simple. Could the POC% vs 14C relationship also reflect a globally high terrestrial (and freshwater?) productivity compared to re-mobilization of petrogenic carbon by erosion? Is biogenic OC supply maybe generally lower in (mountainous) areas of high petrogenic OC supply?

Figure 2: Maybe indicate where C4, C3 and dead carbon would plot in your figures for clarity and easy readability of the graphs?

L. 211: Give potential explanation for these paradox observations.

L.216: Define 'load' - carbon loading is often the POC/SSA ratio, is that what you mean?

L.219: Interesting point. Insert the citations indicated. Maybe in combination with showing variation in POC concentration along depth profile (from database?)?

L.221: Those tropical rivers can still yield high sediment and POC fluxes because of high water discharge, despite relatively low SPM.

L.235: ',' missing.

L.244: They are not only hotspots for sediment production, but also for petrogenic OC mobilization. Also mountainous catchments often expose ancient sediment outcrops providing source for petrogenic OC.

---

## Author Response (AR1)

Dear Editor,

We have revised the manuscript and dataset framework according to four reviewers' comments, furthermore, we updated the MOREPOC database, MOREPOC v1.1 (available on Zenodo: https://doi.org/10.5281/zenodo.7055970) now includes 3,546 SPM data entries, among which 3,053 with POC content, 3,402 with stable carbon isotope ($\delta^{13}$C) values, 2,283 with radiocarbon activity ($\Delta^{14}$C) values, 1,936 with total nitrogen content. This represents a big update compared to MOREPOC v1.0. Aside from the reviewers' comments, we also revised other parts of the manuscript with a focus on strengthening Section 3 "Global riverine POC patterns" and updating all numbers and figures. As also concluded in the manuscript, we hope MOREPOC can benefit our scientific community to understand POC and the bounded carbon cycle.

All changes are marked in the tracking mode in the submitted marked version of the manuscript, and all our replies to reviewers' comments are marked in red as below.

Thank you for your time on this manuscript.

Best regards,

Yutian KE

On behalf of all co-authors

October 13, 2022

Pasadena

**Reviewer 1**

This dataset is a comprehensive inventory of TOC, $\delta^{13}$C, F$^{14}$C, C:N ratios, Al:Si ratios, and important methods related metadata for particulate organic carbon collected and analyzed from rivers around the world. This builds significantly upon previously compiled datasets, whichhave an order of magnitude fewer data points that this new dataset submitted by Ke and colleagues. The publication of this dataset is timely, as the number of studies measuring the geochemistry of fluvial POC has increased over the past decade, and more studies are adopting the dual isotope measurement approach. I am excited about the future studies this dataset will enable.

This dataset seems to be thorough with respect to including all available published data and the dataset includes the relevant parameters and metadata needed to understand how thedata were collected. However, there are some issues with the dataset, particularly with respectto reporting measurement uncertainties, naming the variables used to represent the data, and formatting and populating sampling dates. There are also minor issues throughout the text thatneed to be addressed before this manuscript can be published.

After the dataset and manuscript have been revised to address all issues detailed below,I support publication of this manuscript in *Earth System Science Data*.

Dear Reviewer 1,

Thanks for the detailed comments, which really helped to improve the MOREPOC dataset and the manuscript. We have made several modifications to MOREPOC v1.0, and now version 1.1 is available on Zenodo. Besides, we have revised the manuscript based on the major issues and detailed items addressed in the comments. Please check our reply to the comments below (marked in red):

**Detailed comments:**

*Dataset:*

One significant issue with this dataset is that there are no uncertainties reported for the POC $^{13}$C, F$^{14}$C, and radiocarbon age measurements. Analytical uncertainties are typically requiredwhen publishing these isotopic measurements, so they should be available in most publisheddatasets. Please add these as columns in the dataset.

Indeed, uncertainties are important, thanks for pointing this out. we looked through all references from MOREPOC v1.1 to add analytical uncertainties if available. However, most papers do not report uncertainties. "perc_poc_1sd", "d13C_1sd", and " D14C_1sd" columns were added to provide analytical uncertainties for POC content, $\delta^{13}$C and $\Delta^{14}$C data, respectively.

The names of the first two columns of the dataset (riv_na and bas_na) are not intuitive. Without looking carefully, I would interpret riv_na to be the name of the sampled river, which bas_na would be the name of the major drainage basin that river is in. To ease use of the dataset, I recommend switching these column headers so that riv_na is the name of the sampled river and bas_na is the name of the larger drainage basin to which the sampled river contributes.

To resolve the confusion, the names of some fields were changed to make them more intuitive, e.g., "riv_na" was changed to "bas-id" to refer to the sampled drainage basin and "bas_na" was changed to "riv_id" to refer the name of the sampled river.

With respect to the river name, I suggest removing the country name from the river name andadding a separate column for the country. In its current format, if someone wants to filter the dataset by specific river, they need to type in the country in parentheses after the river name,which makes data analysis challenging.

It is a good point, The "country" column was created to separate it from "riv_id".

CN_mar and as_mar also are not intuitive variable names. CN_ratio and alsi_ratio seem more appropriate and do not exceed the maximum number of characters used for other variables inthe dataset. Rca could also be changed to age_14C.

"RCA" was changed to "age_14C", and "cn_mar" and "as_mar" were changed to "cn_ratio" and "alsi_ratio", respectively.

91 samples do not have dates associated with them, which I assume is because the dates werenot reported in the publication. I would appreciate if the authors could confirm whether they double checked the data sources for information on sampling dates. 549 of the reported measurements do not have individual dates, but a range of years (e.g., 1999-2004). This is not very useful for someone who wants to calibrate POC data to points on river hydrographs. It is also unclear whether the reported geochemical measurements are from individual samples, or a calculated composite measurement of multiple samples. This either needs to be explained in the text, or the dataset needs to be edited to breakout the samples into individual dates.

We added more sampling date information, such as those from Bouchez et al., (2014). Now there are only 3 samples without associated dates. However, some data entries only have the sampling date information as a period (year) because there is no specific date reported for each data entry, such as for a large amount of data from Taiwan rivers (Hilton et al., 2010). Besides, we also addressed that each data entry is an individually collected sample unless several size fractions were reported in the original study in the manuscript.

One of the samples (Amazon River at Obidos, 2005) has the latitude and longitude reversed, such that it plots down near Antarctica. It appears that this point was removed from Figure 1,but it is still in the spreadsheet and in the shapefile.

We corrected this error in all files of different formats, and please note that an average value is presented when several samples have been collected at the same location in Figure 1.

*Manuscript text:*

Line 7-8: This sentence could be re-organized to digest more easily. I suggest "Riverine transport of particulate organic carbon (POC) associated with terrigenous solids to the ocean has an important role in the global carbon cycle."

It was rephrased to "Riverine transport of particulate organic carbon (POC) associated with terrigenous solids to the ocean has an important role in the global carbon cycle".

Line 31: "…over geological timescales (>100,000 years)."

Corrected

Line 45: While SOC in permafrost regions is certainly depleted in 14C because it is old and 14C still decays over time, this organic matter may not have a long turnover time. When frozen, its decomposition rate is zero, but when thawed it may actually have a fast turnover time. We don't yet have enough data to constrain the turnover time because the turnover time

"clock" isonly set once the permafrost thaws, which has been occurring more recently.

We agree that SOC in permafrost has a dynamical turnover time subjected to varying climate conditions, especially the current elevation of ambient temperature will accelerate the thawing of permafrost. In the manuscript, we now explain the term "turnover time" following Eglinton et al. (2021), which is the ratio of soil carbon stock to input flux, to make it more proper to discuss the relative long SOC turnover time in permafrost regions.

Line 47: The Carvalhais et al. (2014) study shows total ecosystem carbon turnover times, not necessarily soil organic carbon turnover time and is not constrained by 14C data, so using thisreference is a misleading.  The Shi et al. (2020) reference is appropriate here.

Carvalhais et al. (2014) was removed from the Introduction.

Line 53: Write "in reservoirs" rather than "at reservoirs."

Replaced.

Line 66: Suggested edit: "…with 531 reported d14C measurements."

Rephrased.

Line 81: "Decarbonization" is not an appropriate word to use here. I suggest "carbonate removal methods." It is also unclear what is meant by "acid adopted." Do you mean type ofacid used for removing carbonate? If so, please reword for clarity.

We agree that "carbonate removal" is more appropriate and is now used to replace "decarbonization" in the manuscript. "acid adopted" means the type of acid used for removing carbonate - this is clarified now.

Table 2:

- Decarbonization is not an appropriate term to use here, because it implies removing allcarbon. Rather, the authors should use "carbonate removal."

Line 126: Again, not sure that decarbonization is the right term to use to describe carbonate removal. It implies that all carbon is removed from the samples. It is more straightforward to say "carbonate removal method" or "inorganic carbon removal method."

We used "carbonate removal" to replace decarbonization in the manuscript.

Lines 134-136: What are the units associated with time and temperature of acid treatment?This is also no reported in the data spreadsheet.

Time is in units of hours and temperature is in units of Celsius degrees. We added this information in the manuscript to explain the parameter in MOREPOC v1.1_RM.

Line 142: "…consists *of* an extensive dataset…"

Rephrased.

Line 146: There should be a comma or colon after 1950, not a quotation mark.

Corrected.

Lines 142-154: Both of these equations have F14C as an input, but the authors never provide anequation to derive F14C from an AMS measurements of $^{14}C/^{12}C$. I recommend adding the equation for $F^{14}C$: Where the denominator is 95% of the 14C activity of the Oxalic Acid 1 standard material in 1950, and the numerator is corrected for fractionation to a common d13C value of -25 per mil.

We agree that the definition of $F^{14}C$ should be in included in the manuscript - the equation was added.

Line 170: "range" not "ranges"

Corrected.

Line 259: Are Earth system modelers not a part of the scientific community? This sentence needs to be re-arranged or re-worded. I suggest deleting "as well as Earth system modelers." It would be worth adding a sentence mentioning that having such a dataset to work with can helpinform and validate Earth system models, improving our ability to model the global carbon cycle.

Good suggestion, we rephrased the expression to "MOREPOC will benefit the scientific community carrying out research on riverine POC sources, transport, and fate. Furthermore, it will help feed and validate Earth system models in order to improve the ability of models to constrain all the components of the global carbon cycle."

**Reviewer 2**

Ke and colleagues provide a large (~ 4x the size previously compiled sets) compilation of published data on riverine particulate organic carbon (POC, incl. isotopic composition and related N content), suspended particulate matter concentrations (SPM),and Al/Si weight ratios of the corresponding sediment. This comprehensible dataset is f good quality and accompanied by a wealth of metadata,such as geographical or methodological information, improving its interpretability and usability of the database. However, uncertainties are commonly reported alongside carbon isotopic values andcould be integrated into the database. Clarity and variable naming could also be improved.

Otherwise, I have only a few minor comments regarding the database (see below).

The descriptive article adequately summarizes database content, structure and patterns within the data. It gives most background information necessary to understand relevance, quality and acquisition of the data. At times the article is written too much in a POC-expert language and missesa few explanations necessary to fully understand the data. More specific comments are attached.

This publication seems timely, relevant and useful to the Earth science community and, generally, researchers interested in riverine and coastal organic matter processes and carbon cycling. The sizeand high spatial coverage of the set provide a proper statistical basis and will certainly help improving our understanding of terrestrial and marine carbon cycling. Detailed comments on data and article can be found below. After these issues have been addressed, I strongly support publication in Earth System Science Data.

Dear Reviewer 2,

Thanks for the detailed comments and recommendations. We hope MOREPOC can benefit the Earth Science community, in particular, researchers interested in understanding OC-related Earth surface processes and carbon cycling at different time- and spatial scales. Now the compilation includes more data entries with 265 more radiocarbon activity ($\Delta^{14}C$) values, providing exciting observations.

We provide our reply to your comments below.

**Data Comments**

*General*

- consider adding readme with variable & unit information. Consider providing here also values and equations used.

A readme file with information on variables and units was added to the database and is now available on Zenodo too. Equations for parameter definition and conversion were included.

- some variable names could be more intuitive (riv_na, bas_na)

These variable names were changed to a more intuitive term ("riv_na" to "bas_id", "bas_na" to "riv_id").

- Could uncertainties be added, at least of the carbon isotopic values, where uncertainty is usuallyreported? I acknowledge that this is (unfortunately) not really common practice in geochemical edatabase work, but would improve the interpretability (and thus value) of the database significantly.

Uncertainties were added for elemental and isotopic carbon values if available. "perc_poc_1sd", "d13C_1sd", and "D14C_1sd" columns were added to provide analytical uncertainties for POC content, $\delta^{13}C$ and $\Delta^{14}C$ data, respectively. Please also check the reply to review 1 for the same comment.

*In MOREPOC v1.0_RM*

- add which variables given in which paper (apart from 14C & 13C)?

We added extra fields to clarify which variable was given in which paper. The columns "para_m" and "para_c" were added to clarify the sources of parameters in MOREPOC v1.1: data entries of some parameters were directly taken from the cited references, while some were obtained through later calculations and conversions.

*In MOREPOC v1.0*

- Origin of values is not always clear: e.g., SPM of Santa Clara in Masiello & Druffel (2001) is derived from USGS monitoring & extrapolation based on relationship of SPM and water discharge. Couldn't find the actual numbers given in the database within the paper by Masiello & Duffel, except for 1 sample (March 25, 1998 with 17,813 mg/L in database and 17.,230 mg/L in M&D paper). How comesthere is a difference? Did you use POC conc. and POC wt% to derive SPM conc.? Please indicate whether individual variables are originally given in the corresponding paper and/or more detail on how variables were derived.

Referring to the reply to the previous comment, we added two columns to clarify the sources of the compiled data. For references only reporting POC wt.% and concentration, we did use the two parameters to derive SPM concentration. Parameters obtained from this conversion were clarified. Now, this information is clarified in *MOREPOC v1.1_RM*.

- some variables could be formatted more user friendly ()

- RCA '0' instead of 'modern' too keep it numerical? Also '14C_age' or similar could be amore intuitive name

- No brackets, points, commas or empty spaces in the data fields, especially in text 'values'(or strings)

- Separate countries and continent from 'cont' variable.

For these four comments on format problems: We did not change "modern" to "0", since some high $^{14}C$ activity data are because of the influence of the bomb carbon effect. As a consequence, we would like to have users decide how to use these values themselves. However, we changed the field name as suggested. For 'values' in data fields, we converted strings to numeric format and eliminated the use of brackets, points, commas, or empty spaces. Country information is now included in a new column "country".

**Article CommentsGeneral:**

The text is well structured and easily readable. It gives a good background and basic understanding to the database published. However, it could be written in less expert-language, could better explainnotations/values and citations could be more thorough/original.

We revised the manuscript to make the writing a less expert language, please check it.

**Specific:**

L21: Not only input but also susceptibility to mineralization and specific environment where itsdeposited are key features of carbon cycling (see Blair & Aller 2012, who are cited just before).

We added a sentence on the role of alteration and degradation on fluvial POC goes during transport in terrestrial environment before entering the coastal environment and being buried in the ocean.

L24: The 'biogenic' source could be split into river-authigenic, land plant (litterfall) and soil organiccarbon. Generally, riverine authigenic POC generally seems to be a little underappreciated in the article.

We clarified this statement about the composition of biospheric OC. Riverine authigenic POC is now mentioned to partly explain the depleted $^{13}$C signals in riverine POC (section 3.1).

L32: Are you referring global carbon fluxes of biogenic POC sequestration? Clarify and give estimatesof these fluxes?

We now refer to the global POC$_{bio}$ burial in the ocean, as well as to the oxidation of POC$_{petro}$. The terrestrial POC$_{bio}$ burial can be up to around 70 MtC/yr considering an average burial efficiency of 30% to an input of ~110-230 MtC/yr (Blair and Aller, 2012; Burdige, 2005; Galy et al, 2015), while the oxidation of POC$_{petro}$ in sedimentary rocks can contribute ~40-100 MtC/yr to atmospheric $CO_2$ (Petsch, 2014; Hilton and West, 2020).

L35: 'played by' could be 'of' to be slightly more concise.

Rephrased.

L47-48: Evtl. highlight here the contribution of old (petrogenic) carbon release from thawing permafrost?

We highlighted that the input of aged biospheric OC from thawing permafrost is the major reason (Wild et al., 2019; Hilton et al., 2015). Besides, we extended the discussion on permafrost-derived fluvial POC in section 3 (in particular section 3.2) and figures 6 and 7.

L.50-51: It is also because of sedimentary dynamics (e.g., steady accumulation scenario vs. fluid andmobile mud layers) and increased oxygen exposure in highly energetic systems (the 'incinerator' concept of Blair & Aller).

We agree that sediment dynamics and oxygen availability in marine environments are important factors (Blair and Aller, 2012). However, in this paragraph, we want to discuss how POC can be altered in terrestrial settings before entering the marine environment. We do emphasize the importance of sediment dynamics in the terrestrial environment due to the different tectonic settings.

L.54: Add reference: Dethier et al. 2022 Rapid changes to global river suspended sediment flux by humans. Science 376(6600), pp.1447-1452. There are also considerations on the impact of humanson riverine carbon cycling (by Taylor Maavara et al. 2017, Nature, or van Hoek et al. 2022, ES&T, among others). It would be suitable for the paper to address this here.

Reference added. Besides, we also added some clarification on how human activities influence erosion and on the resulting changes in the fluxes of sediments and associated POC.

L61-62: Water quality datasets have significantly improved since 1996. This should be acknowledgedhere. There are also more recent and increasingly sophisticated model of riverine carbon cycling andfluxes (summarized e.g., in van Hoek et al. 2022, ES&T). These should also be acknowledged here.

We added a sentence on the improvement of water quality datasets and increasingly sophisticated models of riverine carbon cycling as you suggested.

L.65: Generally it would help the reader, especially if non-expert in respect to carbon isotopes, ifnotations (here D14C) and units (section 2.7) could be explained before using them in the text. Fm14C is never explained, not even in section 2.7, but extensively used.

We added more explanation for D14C and Fm14C in section 2.7.

L.79: Which statistical examinations were used?

This is based on non-numeric string detection in value fields, categorical summarization, and extreme numbers detection. We were wrong here to use the term – statistical examinations, so we deleted this expression.

L155: If space allows it may be instructive to mention Al/Si can also be related to specific surfacearea to derive carbon loading (mass OC per mineral surface area).

We added explanations on why Al/Si ratio is an important parameter that is included in MOREPOC as follows:

"Lastly, if available, the aluminum-to-silicon mass ratio (Al/Si) is also provided in MOREPOC v1.0 1. This elemental ratio is an efficient proxy for the particle size of riverine sediment, allowing to characterize the grain size effect of sediments on POC loading in fluvial delivery (Bouchez et al., 2011; Galy et al., 20072008b; Bouchez et al., 2011; Hilton et al., 2015). The mineralogy and particle size of sediments are generally related, with coarse particles being quartz-rich (low Al/Si ratios) and fine particles being clay-rich (high Al/Si ratios) (Galy et al., 2008b). POC contents are usually positively correlated with proportions of fine-grained fractions (Mayer, 1994; Galy et al., 2008b; Bouchez et al., 2014)."

L.158: Also no data from Greenland and arid regions (N Africa & Arabia) as far as I can see from Fig 1.L.159: If space, add a sentence or two on the other continents?

We added this information "It can be noticed that there are relatively abundant POC studies in North America fluvial systems. However, MOREPOC database also indicates the lack of studies on POC in fluvial systems in the cryosphere regions such as Antarctica and Greenland as well as in arid regions, including Australia, and vast areas spanning from northern Africa to middle east Asia (Figure 1)."

L.161: Maybe add to this suggestion a note on what could be done with this data and what would be the added value for insights into POC processes?

We have included this part in the conclusion for the current dataset.

L.166: There is also control by biology/vegetation and by in-stream and generally fresh-water processes.

These two factors are not external or internal geological forces, and these two processes is regulated by tectonics, climate etc. So we did not add these and other potential factors into this paragraph to explain the control of the mobilization of terrestrial organic matter into fluvial systems.

L178-179: The Fm14C in SE Asia (Fig 1) should also be mentioned here!

We added this information: "Around the Qinghai-Tibet Plateau, where most large river systems in eastern and southern Asia share similar high-elevation headwaters, POC is usually characterized by relatively depleted 14C signals due to strong erosion of sedimentary rocks, such as the Ganges-Brahmaputra (Galy et al., 2007) or the Changjiang (Wang et al., 2012; Wang et al., 2019), and to the erosion of soil, pre-aged OC, e.g. the Huanghe (Tao et al., 2015)."

L.197: This explanation seems a bit too simple. Could the POC% vs 14C relationship also reflect a globally high terrestrial (and freshwater?) productivity compared to re-mobilization of petrogeniccarbon by erosion? Is biogenic OC supply maybe generally lower in (mountainous) areas of high petrogenic OC supply?

Net primary production does not show any clear relation with $^{14}$C in fluvial POC nor with the relative abundance of biospheric OC. It is rather the erosion rate of the catchment which controls the flux of biospheric OC (Galy et al., 2015). We added additional explanations for this trend: "This might reflect the major POC components: 1) dominated by POC$_{bio}$, the combined effects of increasing coverage of C4 plants in tropical regions and the input of pre-aged OC$_{bio}$ of C3 plants from degrading permafrost at high latitude (Cerling et al., 1997; Still et al., 2003); 2) dominated by POC$_{petro}$, rivers in mountainous regions tends to erode $^{13}$C-enrich petrogenic OC (Hilton et al., 2010; Galy et al., 2007). In addition, aquatic authigenic production can be an important mechanism contributing $^{13}$C-depleted and $^{14}$C-enriched POC (Longworth et al., 2007; Marwick et al., 2015; Wu et al., 2018)."

Figure 2: Maybe indicate where C4, C3 and dead carbon would plot in your figures for clarity andeasy readability of the graphs?

We did not add different potential endmembers because we want users to interpret possible sources for fluvial POC.

L. 211: Give potential explanation for these paradox observations.

Potential explanations for this observation were added.

L.216: Define 'load' - carbon loading is often the POC/SSA ratio, is that what you mean?

This occurrence was not about carbon loading, we are sorry about the confusion caused. We were talking about the POC flux that is loaded in a specific SPM concentration. It is now clarified in the text.

L.219: Interesting point. Insert the citations indicated. Maybe in combination with showing variation in POC concentration along depth profile (from database?)?

We added a figure as suggested, providing POC concentration vertical variation in the water column as obtained from depth profiles in large rivers. Selected depth profiles are from the Yukon (Holmes et al., 2022), Mackenzie including Peel and Arctic Red (Hilton et al., 2015), Amazon (Bouchez et al., 2014), and Ganges-Brahmaputra-Meghna systems (Galy et al., 2007, 2008b).

[Figure]

Figure 5: POC concentration variation in vertical water columns from depth profiles in global large rivers. Selected depth profiles are from the Yukon, Lena, and Ob' (Holmes et al., 2022), Mackenzie including Peel and Arctic Red (Hilton et al., 2015), Amazon (Bouchez et al., 2014), Paraná (Repasch et al., 2021), and Ganges-Brahmaputra-Meghna systems (Galy et al., 2007, 2008b). Normalized river depth is calculated by normalizing the individual sample depth to the maximum sample depth of the corresponding profile.

L.221: Those tropical rivers can still yield high sediment and POC fluxes because of high water discharge, despite relatively low SPM.

We found this sentence was not correct under certain circumstances, so we reworded to "Small SPM concentrations (less than 10 mg/L) are generally found in rivers during the frozen season or rivers draining either high-latitude or tropical areas characterized by low-relief settings, in which POC content is relatively high (Gao et al., 2007; Holmes et al., 2022)"

L.235: ',' missing.

Added.

L.244: They are not only hotspots for sediment production, but also for petrogenic OC mobilization. Also mountainous catchments often expose ancient sediment outcrops providing source for petrogenic OC.

Good point, we now refer to petrogenic OC mobilization. However, this paragraph is focused on the erosion of sediments and aims to explain the depleted $^{14}$C nature of fluvial POC observed in some conditions. As a consequence, the role of oxidation of petrogenic OC is not mentioned in the text.

Good point, we now refer to petrogenic OC mobilization. However, this paragraph is focused on the erosion of sediments and aims to explain the depleted $^{14}$C nature of fluvial POC observed in some conditions. As a consequence, the role of oxidation of petrogenic OC is not mentioned in the text.

**Reviewer 3**

The manuscript Ke et al. presents a new openly-accessible global database of organic carbon (OC), OC isotopes ($^{13}$C and $^{14}$C) and key element ratios (Al/Si) in riverine suspended matter entitled "*MOdern River archivEs of Particulate Organic Carbon – MOREPOC*". The database aims to provide data to study OC release, transport and cycling across river-basin systems, which serves the increasingly-important purpose of understating global carbon cycling. The MOREPOC builds on a large number of earlier studies that laid out the ground work and published most of the data that is now curated in this database. Hence, this compilation increases the accessibility and usefulness of already-published work, and harmonizes POC measurements across studies and regions in one easy-understandable data set. Standalone, or along with other data collections from land or ocean, I expect this database to be very useful to facilitate a range of biogeochemical studies and I commend the authors for this effort.

The paper is comprehensible, well-structured and fulfills the purpose of describing the database very well. In their writing, the authors also provide a rough outline of the large-scale differences in fluvial OC concentrations and composition, and provide a short perspective of their interpretations. I only have a few minor comments that are described below, and I recommend publication after the authors have addressed these and the other comments provided by the other reviewers.

Dear Reviewer 3,

Thanks for your comments, which helped to improve the dataset. MOREPOC v1.1 is now available on Zenodo. This refined version of the database now includes 3,546 SPM data entries, among which 3,053 with POC content, 3,402 with stable carbon isotope ($\delta^{13}$C) values, 2,283 with radiocarbon activity ($\Delta^{14}$C) values, 1,936 with total nitrogen content. This represents a significant update compared to MOREPOC v1.0.

Our replies to your comments are as follows:

**Minor points**

Line 26 "radiocarbon-enriched POC": May not apply to soils, deeper soils are more depleted in radiocarbon

line 27: Similar comment like above, if $^{14}$C ages are "multi-millennial" they cannot be enriched

For Line 26-27: We rephrased this sentence in more appropriate terms: " Land plants, soils, aquatic organisms, and microbes can all contribute radiocarbon-active riverine POC, with ages ranging from modern to multi-millennial (Galy et al 2007; Blair et al., 2010; Hilton et al., 2011)".

Line 29 "full erosion/sedimentation/exhumation cycle": If the authors mean rocks would this also include diagenesis and organic carbon maturation processes?

We are here mainly talking about the whole process that terrestrial OC being eroded to final burial in the ocean bottom and then entering the sedimentation cycle, and finally being re-exposed to the atmosphere.

Line 35 "role played by POC": first, I would rephrase this to "the role of POC in the global carbon cycle". Second, POC probably plays only a very small role in the global carbon cycle when compared to other fluxes. However, POC provides

very valuable information about the global carbon cycle as it provides an integrated signal of biogeochemical processes over large drainage-basin areas.

The sentence was rephrased: "Net continental $POC_{bio}$ burial accounts for about 35-70 MtC/yr considering that only 30% of the total riverine input to the ocean is efficiently buried (Blair and Aller, 2012; Burdige, 2005; Galy et al, 2015), while the oxidation of $POC_{petro}$ in sedimentary rocks would release about 40-100 MtC/yr to the atmosphere (Petsch, 2014; Hilton and West, 2020). These fluxes are comparable to those induced by silicate weathering, carbonate weathering by oxidation of sulfides, and volcanism, demonstrating that POC could play an important role in the Earth's long term carbon cycle (Berner, 2003; Hilton et al., 2014; Petsch, 2014; Galy et al., 2007; Galy and Eglinton, 2011; Hilton and West, 2020)."

Introduction: A clear definition of POC, and how it distinguishes from DOC and other OC phases, should be included in my opinion.

We added the definition of POC to the text: " POC is defined as the fraction of total organic carbon contained in the solid fraction recovered after filtration of river water.".

Line 84: It is unclear what the authors mean by "projected in a Geographic Coordinate System". Do the authors mean that the data entries have coordinates according to this coordinate system? Or were the coordinates converted (and re-projected) from one to another coordinate system?

We were referring to the coordinate system used when digitalizing MOREPOC data entries in a shapefile layer in ArcGIS 10.3, We rephrased this sentence: "Location of samples was digitalized if available, and an associate ArcGIS data layer in shapefile format (see MOREPOC_v1.1.rar) is provided with all points projected in a Geographic Coordinate System using the World Geodetic System 1984 (WGS1984)."

Section 2.4: This is all good information but I am missing a description about the sampling location along the course of the river (e.g. river mouth, headwater, center of the river, …). Is this information somehow included in the database (e.g. via the coordinates) or can the authors describe where POC usually is sampled in a river?

This is definitely a good point that will need more work in the future. In such a wide compilation, sampling locations correspond to catchments of different scales, i.e., main channels vs. tributaries as well as sub-tributaries. However, it is challenging to summarize the information on whether samples were collected at a river mouth, along the flowing routing or the headwater. Nevertheless, we believe that looking into the map of MOREPOC sampling points (such as the ArcGIS product provided with MOREPOC) the reader will be able to find that most sampling locations represent integrated signals of biogeochemical processes over a whole catchment.

Line 121: "mesh size" instead of "porosity"?

We changed to use mesh size.

Line 172: This value is for C3 vegetation only – what about C4 plants?

C3 plants make up over 95% of the global biomass, such that it would be hard to find significant C4 signals in global POC patterns. We now explain in section 3.1: "However, POC-rich riverine SPM can also be relatively enriched in $^{13}C$, $i.e.$, $\delta^{13}C$ values larger than -20‰ (Figure 2 and Figure 3). This pattern indicates the presence of an additional pool of $^{14}C$- and $^{13}C$-

rich POC in the terrestrial environment (Cerling et al., 1997), consisting of modern C4-plants in catchments dominated by grasslands or savannah (*e.g.,* Marwick et al., 2015)."

General 3.1: Could $^{13}$C values also be affected by degradation of POC during fluvial transport, and thus affect the isotopic source signal? Some of the values (e.g. <-30‰) are outside the typical window of plant OC.

$\delta^{13}$C values can be affected by the degradation of POC during fluvial transport, as shown by Mayorga et al. (2005), with the preferential degradation of young, labile biospheric OC resulting in an increase of $\delta^{13}$C values. However, such effect typically does not result in $\delta^{13}$C values outside the range of C3 plant OC. We added an explanation for the very $^{13}$C-depleted signals in fluvial POC in Section 3.1. Actually, these samples are mostly from permafrost-draining rivers; and secondly, aquatic authigenic OC production can be an important mechanism contributing $^{13}$C-depleted and $^{14}$C-enriched POC (Longworth et al., 2007; Marwick et al., 2015; Wu et al., 2018).

Line 257: Have the Al/Si ratios been introduced and described somewhere? I suggest to include a brief description in the introduction. Al/Si ratios and its purpose for river-based investigations may not be obvious to all readers.

Al/Si was not introduced properly in previous version of the manuscript. We now explain in section 2.7 why MOREPOC features Al/Si data: "Lastly, if available, the aluminum-to-silicon mass ratio (Al/Si) is also provided in MOREPOC v1.0 1. This elemental ratio is an efficient proxy for the particle size of riverine sediment, allowing to characterize the grain size effect of sediments on POC loading in fluvial delivery (Bouchez et al., 2011; Galy et al., 20072008b; Bouchez et al., 2011; Hilton et al., 2015). The mineralogy and particle size of sediments are generally related, with coarse particles being quartz-rich (low Al/Si ratios) and fine particles being clay-rich (high Al/Si ratios) (Galy et al., 2008b). POC contents are usually positively correlated with proportions of fine-grained fractions (Mayer, 1994; Galy et al., 2008b; Bouchez et al., 2014)."

**Reviewer 4 Jin Wang**

In this paper, Ke and co-workers compiled a global database of river POC, including the carbon isotopes, nitrogen content and aluminum-to-silicon ratios. Using this dataset, the authors showed the pattern of stable carbon and radiocarbon isotopic ratios of the global river POC and investigated the controls on the source of the POC. The dataset of this paper is very useful for the studies of river organic carbon and therefore, is useful for global carbon cycle and modelling.

The dataset is generally well formatted, and this paper is well-written. It would be interested a broad biogeochemistry community. I only have some minor questions, which I hope are useful for the authors to strength the paper.

Dear Dr. Jin Wang,

Thanks for the comments. We have revised the manuscript accordingly.

MOREPOC v1.1 is an updated version of MOREPOC v1.0, both available on Zenodo. The number of data entries increases to an amount of 3,546 during revision, among which 3,053 with POC content, 3,402 with stable carbon isotope ($\delta^{13}$C) values, 2,283 with radiocarbon activity ($\Delta^{14}$C) values, 1,936 with total nitrogen content.

Our replies to your comments are as follows:

**General comment:**

1. Grain size is a very important factor controlling the characteristics and fate of the POC. Since the Al/Si ratio has been compiled in the dataset, the paper could have some text to address the controlling of grain size on the concentration, source, and characteristics of the POC.

We realized that there was only a poor introduction to the interest of the Al/Si ratio in the context of POC studies, a parameter included in MOREPOC v1.1. We added to section 2.7: "Lastly, if available, the aluminum-to-silicon mass ratio (Al/Si) is also provided in MOREPOC v1.0 1. This elemental ratio is an efficient proxy for the particle size of riverine sediment, allowing to characterize the grain size effect of sediments on POC loading in fluvial delivery (Bouchez et al., 2011; Galy et al., 20072008b; Bouchez et al., 2011; Hilton et al., 2015). The mineralogy and particle size of sediments are generally related, with coarse particles being quartz-rich (low Al/Si ratios) and fine particles being clay-rich (high Al/Si ratios) (Galy et al., 2008b). POC contents are usually positively correlated with proportions of fine-grained fractions (Mayer, 1994; Galy et al., 2008b; Bouchez et al., 2014)."

2. the expression of POC concentration is confusing. I see the authors try to separate POC in the unit % and mg/L, using POC content for the unit wt.% and using POC concentration for the unit mg/L. but there is still someplace confusing, e.g., Line 210. Please clarify them in the text.

This confusion has been resolved in the revised manuscript: in Line 210, we changed the term "POC concentration" to "POC content".

3. please consider adding POC in wt.% versus SPM into Fig. 4 as panel B. I guess there would be a dilution trend.

We added a color bar (for POC wt.%) in Figure. 4 to indicate the dilution of organic carbon by inorganic materials. The constant-POC countour lines drawn in the figure also provide the same information.

Specific comments in the text:

Line 47: need to specify that the "riverine POC" is "riverine $POC_{bio}$". The conclusion is not right if taken the petrogenic OC in account. For instance, Taiwan rivers have high fraction of petrogenic carbon, thus very old total POC.

We changed to "riverine $POC_{bio}$" in the text.

Line 78: Why error could be generated during the compilation? I guess some papers only show data on the figure, how did you convert them to values?

In some references, data tables were indeed images, such that we had to use automatic conversion tools to transfer image data into tables. However, this may result in some numbers being converted into letters, which we had to carefully check. We should have not taken any data from figures that need to use Digitalizer to take the approximate number, we sent emails to corresponding authors to inquire data.

Line 129: There is another decarbonization method that has been used in some papers. Carbonate is in-situ removed by adding liquid HCl in silver capsule, and then oven-dried (Menges et al., 2020, GCA). Also, I found this paper is missed in the data compilation.

Line 129: Reference to this carbonate removal method was added to the manuscript, although it seems more common in soil OC studies than riverine POC studies. We also added the data from Menges et al. (2020) to MOREPOC v1.1 (section 2.6).

Line 201-205: The argument should be careful here. First, I don't see a very clear trend. Second, the δ13C and Δ14C of the POC is generally controlled by the fraction of petrogenic versus biospheric POC in the global rivers (Fig. 2). Therefore, the high or low δ13C and Δ14C are more related to the fraction of different endmembers (including the C3, rock and in situ production beside C4 plants). I found that the Taiwan rivers and Congo rivers both have high δ13C, but the reason is different, the former is because of high contribution from rock.

We provided extra explanations to strengthen the statement: "This might reflect the major POC components: 1) dominated by $POC_{bio}$, the combined effects of increasing coverage of C4 plants in tropical regions and the input of pre-aged $OC_{bio}$ of C3 plants from degrading permafrost at high latitude (Cerling et al., 1997; Still et al., 2003); 2) dominated by $POC_{petro}$, rivers in mountainous regions tends to erode [13]C-enrich petrogenic OC (Hilton et al., 2010; Galy et al., 2007)."

Line 220: reference Hilton et al. (2008) studied the impact of typhoon on the POC source and flux. Please consider adding the reference Wang et al. (2015, Geology) or Firth et al. (2018, Nature Geoscience) for referring to the impact of earthquake.

We added the suggested references.

Line 222: This sentence may be not correct. Taiwan, Amazon and Ganges are tropical rivers, but are very high in SPM and the POC content is not high in Taiwan either.

We realized that there was a problem with this sentence, which we reworded to "Small SPM concentrations (less than 10 mg/L) are generally found in rivers during the frozen season or rivers draining either high-latitude or tropical areas characterized by low-relief settings, in which POC content is relatively high (Gao et al., 2007; Holmes et al., 2022)"

Reference: some references are missing, e.g., Wang et al., 2012, Wang et al., 2019.

Missing references were added to the Reference section.